# Mycobacterium tuberculosis canonical virulence factors interfere with a late component of the TLR2 response

Amelia E Hinman[1], Charul Jani[1], Stephanie C Pringle[1], Wei R Zhang[1], Neharika Jain[2], Amanda J Martinot[2], Amy K Barczak[1,3,4]*

[1]The Ragon Institute, Massachusetts General Hospital, Cambridge, United States; [2]Department of Infectious Diseases and Global Health, Tufts University Cummings School of Veterinary Medicine, North Grafton, MA, United States; [3]The Division of Infectious Diseases, Massachusetts General Hospital, Boston, United States; [4]Department of Medicine, Harvard Medical School, Boston, United States

**Abstract** For many intracellular pathogens, the phagosome is the site of events and interactions that shape infection outcome. Phagosomal membrane damage, in particular, is proposed to benefit invading pathogens. To define the innate immune consequences of this damage, we profiled macrophage transcriptional responses to wild-type *Mycobacterium tuberculosis* (Mtb) and mutants that fail to damage the phagosomal membrane. We identified a set of genes with enhanced expression in response to the mutants. These genes represented a late component of the TLR2-dependent transcriptional response to Mtb, distinct from an earlier component that included *Tnf*. Expression of the later component was inherent to TLR2 activation, dependent upon endosomal uptake, and enhanced by phagosome acidification. Canonical Mtb virulence factors that contribute to phagosomal membrane damage blunted phagosome acidification and undermined the endosome-specific response. Profiling cell survival and bacterial growth in macrophages demonstrated that the attenuation of these mutants is partially dependent upon TLR2. Further, TLR2 contributed to the attenuated phenotype of one of these mutants in a murine model of infection. These results demonstrate two distinct components of the TLR2 response and identify a component dependent upon endosomal uptake as a point where pathogenic bacteria interfere with the generation of effective inflammation. This interference promotes tuberculosis (TB) pathogenesis in both macrophage and murine infection models.

*For correspondence:
ABARCZAK@mgh.harvard.edu

**Competing interest:** The authors declare that no competing interests exist.

## Editor's evaluation

This article provides insight into the kinetics of TLR2 mediated immune responses and the roles for Esx1 and PDIM in these responses. These results demonstrate two distinct components of the TLR2 response and show the relevance of these responses to *Mycobacterium tuberculosis* pathogenesis. These findings are important and of interest to the broader field of host cell interactions with intracellular pathogens.

## Introduction

Innate immune recognition of invading pathogens, typically driven by the interaction of pattern recognition receptors (PRRs) and pathogen-associated molecular patterns (PAMPs), requires recognition of microbial products at multiple subcellular sites. While some PRRs recognize PAMPs at a single site within the cell, other PRRs have the potential to bind PAMPs and initiate signaling from

multiple sites. The mechanisms through which one PRR can recognize and respond distinctly to PAMPs at different subcellular sites is best understood for TLR4/LPS interactions (*Bonham et al., 2014*; *Fitzgerald et al., 2003*; *Kagan and Medzhitov, 2006*; *Kagan et al., 2008*; *Yamamoto et al., 2003*). Although principles elucidated with LPS and TLR4 are broadly thought to hold for other PAMP/PRR interactions, to date we have less insight into subcellular sites of signaling by other PRRs, the contribution of compartment-specific signaling in response to complex micro-organisms, and the pathogenic strategies employed to evade such compartmentalized signaling events.

TLR2, a receptor for bacterial cell wall lipoproteins, has been suggested to signal from the plasma membrane and endosomes, similar to TLR4. Endosome-specific TLR2 signaling in response to pathogenic bacteria has been partially explored using the model of *Staphylococcus aureus* taken up into macrophages (*Ip et al., 2010*); in that work, TNF release was shown to be partially dependent upon TLR2 and dependent upon endosomal uptake. TLR2 activation has also been described to induce a type I interferon (IFN) transcriptional response from endosomes (*Barbalat et al., 2009*; *Dietrich et al., 2010*; *Musilova et al., 2019*; *Stack et al., 2014*). Overall, the mechanisms and physiological contexts in which compartment-specific TLR2 signaling occurs are unclear. In particular, it is unknown whether findings with *S. aureus* extend to other infectious agents and whether pathogens use strategies to prevent TLR2 signaling from the plasma membrane or endosomes.

*Mycobacterium tuberculosis* (Mtb) represents a model to study potential mechanisms of innate immune evasion, as this pathogen co-evolved with mammals and encodes multiple strategies of host manipulation. Mtb has a complex repertoire of PAMPs, and infection with Mtb is recognized by both membrane-bound and cytosolic PRRs. The specific complement of PRRs that drive the macrophage response to the intact bacterium and the subcellular sites of recognition of those PAMPs have not been clearly defined. The canonical Mtb virulence factors phthiocerol dimycocerosate (PDIM) and ESX-1 contribute to disruption of the macrophage phagosomal membrane upon infection (*Augenstreich et al., 2017*; *Barczak et al., 2017*; *Manzanillo et al., 2012*; *Quigley et al., 2017*; *Simeone et al., 2012*); we sought to leverage this shared pathogenic effect to gain insight into compartment-specific signaling in the macrophage response to Mtb.

To probe the relationship between Mtb-mediated phagosomal membrane damage and innate immune recognition of infection, we serially profiled the macrophage response to wild-type Mtb or PDIM and ESX-1 Mtb mutants, which fail to damage the phagosomal membrane. We found that the mutants elicited markedly enhanced expression of a cluster of inflammatory genes induced late after infection; expression was strictly dependent upon MYD88 and TLR2. TNF expression and release are commonly used as a marker of TLR activation; however, we found that induction of *Tnf* occurred with an earlier set of TLR2-dependent genes and differed minimally between the response to wild-type Mtb and our mutants. We thus hypothesized that infection with Mtb elicits a two-component TLR2-dependent response, and that the later component of the response is preferentially blunted by Mtb factors that damage the phagosomal membrane. Treatment of macrophages with synthetic TLR2 ligand elicited a similar two-component transcriptional response, suggesting that these components are fundamental facets of TLR2 signaling rather than pathogen-specific. Induction of the early component of the TLR2 response was similar in the presence of endosomal uptake inhibitors; in contrast, the later component was markedly diminished by inhibition of endosomal uptake. Induction of the endosome-specific response was dependent upon phagosome acidification. We found that Mtb factors known to damage the phagosomal membrane contributed to Mtb-induced limitation of phagosome acidification, which in turn limited production of the late component of the TLR2 response. Consistent with published reports, PDIM-mutant and ESX-1-mutant Mtb had attenuated virulence phenotypes in wild-type macrophages, with reduced macrophage cytotoxicity and reduced bacterial growth. Both of these attenuated phenotypes were partially reversed in *Tlr2-/-* macrophages, suggesting that TLR2-dependent responses contribute to the attenuation of these mutants. As expected, PDIM-mutant Mtb had attenuated infection phenotypes in C57BL/6J mice, with reduced bacterial growth and lung infiltrates. In contrast, in *Tlr-/-* mice PDIM-mutant Mtb grew more robustly and caused pulmonary infiltrates more similar to wild-type Mtb. These results support a model in which PDIM and ESX-1 contribute to Mtb virulence in part by blunting a protective endosome-specific component of the TLR2-dependent response to infection.

## Results

### Macrophage infection with Mtb PDIM or ESX-1 mutants elicits enhanced expression of an inflammatory transcriptional program

The Mtb ESX-1 protein secretion system has long been known to mediate phagosomal membrane damage (*Manzanillo et al., 2012*; *Simeone et al., 2012*); the mycobacterial lipid PDIM was more recently found to play a similar role in infection (*Augenstreich et al., 2017*; *Barczak et al., 2017*; *Quigley et al., 2017*). To understand how the capacity to damage the phagosomal membrane shapes the innate immune response to Mtb, we sought to leverage the similar effects of PDIM and ESX-1-mediated secretion within the macrophage. We thus compared the response to wild-type Mtb with the response to PDIM and ESX-1 mutants, with the goal of identifying facets of the macrophage response to Mtb impacted by phagosomal membrane damage. To enable meaningful comparison, we first tested whether wild-type Mtb, PDIM mutants, and ESX-1 mutants were taken up similarly into bone marrow-derived macrophages (BMDM). While the possibility of PDIM facilitating uptake into macrophages has been raised (*Astarie-Dequeker et al., 2009*; *Augenstreich et al., 2019*), using either CFU or flow cytometry we found that wild-type Mtb strain H37Rv, PDIM mutants, and ESX-1 mutants were taken up at similar rates (*Figure 1—figure supplement 1A–B*).

To define the macrophage transcriptional programs induced by Mtb, we infected BMDM with wild-type Mtb and performed comprehensive transcriptional profiling at 4, 8, 12, and 16 hr post-infection (*Supplementary file 1*). Focusing on the 907 genes changed two-fold upon infection, genes could be categorized into three clusters with distinct patterns of expression (*Figure 1A*). Cluster 1 genes were progressively induced, and cluster 2 genes were progressively repressed over time after infection. Cluster 3 genes were induced upon infection and peaked at 8–12 hr before waning. To identify the facets of the macrophage response most impacted by phagosomal membrane damage, we next compared this baseline response to wild-type Mtb with the response to PDIM or ESX-1 mutants (*Supplementary file 1*). For this comparison, we used PDIM mutants we had demonstrated in previous work to lack PDIM production (*mas* and *ppsD*) and an ESX-1 mutant we had demonstrated lacked ESX-1 secretion but produced and properly localized PDIM (*Tn::eccCa1*) (*Barczak et al., 2017*). Comparing the macrophage response to our mutants with the response to wild-type Mtb, we found that expression of genes in clusters 2 and 3 was similar in response to each of the three strains. In contrast, expression of genes in cluster 1 was markedly impacted by loss of PDIM or ESX-1. However, not all genes in the cluster responded similarly to the mutants; classifying genes in this cluster based on their response to PDIM and ESX-1 mutants clearly distinguished two subclusters (*Figure 1B*). Induction of genes in subcluster 1A was markedly diminished in response to PDIM and ESX-1 mutants relative to wild-type Mtb (*Figure 1B*). Ingenuity pathway analysis (*Krämer et al., 2014*) of genes in this subcluster predicted STAT1, IRF3, IFN, and IFNAR as upstream regulators with high confidence, suggesting that these genes comprised the type I IFN response. Manual inspection confirmed that IFN stimulated genes were highly represented in subcluster 1A. These results were consistent with previous work demonstrating that the macrophage type I IFN response to Mtb is dependent upon ESX-1-mediated secretion (*Stanley et al., 2007*) and PDIM (*Barczak et al., 2017*).

In contrast to subcluster 1A, expression of genes in subcluster 1B was enhanced in response to PDIM or ESX-1 mutants relative to wild-type Mtb at later timepoints (*Figures 1C–D and 2A*). Strikingly, this subcluster included multiple genes important for the host response to Mtb, including *Marco* (*Bowdish et al., 2013*), prostaglandin E synthase (*Chen et al., 2008*; *Garg et al., 2008*; *Mayer-Barber et al., 2014*; *Rangel Moreno et al., 2002*), lipocalin 2 (*Guglani et al., 2012*; *Saiga et al., 2008*), *Irg1* (*Hoffmann et al., 2019*; *Nair et al., 2018*), and matrix metalloproteinase 14 (*Sathyamoorthy et al., 2015*). The kinetics of overall induction and enhanced expression observed in response to the PDIM and ESX-1 mutants were independent of MOI, as the same patterns were observed infecting with MOI 2:1 and 10:1 (*Figure 2B*). Enhanced expression was similarly elicited in BALB/c BMDM, suggesting that the enhanced response is independent of the background genetic inflammatory state of the cells (*Figure 2—figure supplement 1A*). Enhanced expression was observed regardless of the specific PDIM or ESX-1 mutation, including ESX-1 core complex mutant *Tn::eccCa1*, secreted effector mutants (*esxB* and *Tn::espC*), mutants in the synthetic pathway for distinct components of PDIM (*mas* and *ppsD*), and a PDIM transport mutant (*Tn::drrC*) (*Figure 2C*). Complementation of the disrupted ESX-1 or PDIM gene restored expression to wild-type levels (*Figure 2D*). PDIM and ESX-1 mutants are both known to be attenuated for growth in macrophages (*Camacho et al., 1999*; *Stanley et al.,*

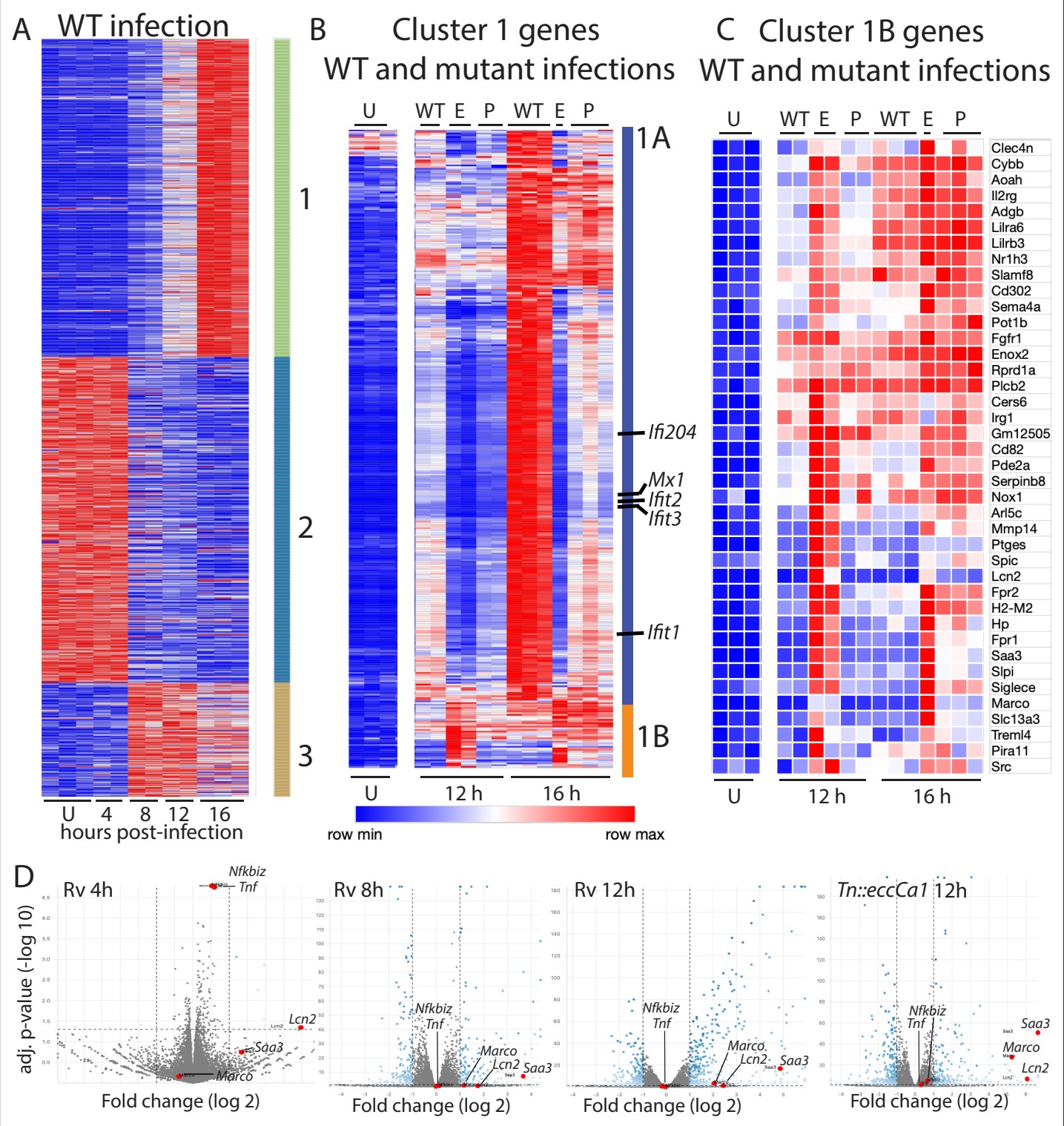

**Figure 1.** Macrophage infection with *Mycobacterium tuberculosis* (Mtb) phthiocerol dimycocerosate (PDIM) or ESX-1 mutants reveals two subclusters of genes differentially expressed relative to infection with wild-type Mtb. (**A–C**) C57BL/6J bone marrow-derived macrophages (BMDM) were infected with wild-type Mtb H37Rv ('WT'), the ESX-1 core complex mutant *Tn::eccCa1* ('E'), or the PDIM production mutant *mas* ('P') at an MOI of 2:1. At 4, 8, 12, and 16 hr post-infection, RNA was harvested for RNAseq. Sequencing libraries not passing QC metrics were excluded from further analysis. (**A**) Genes were clustered based on similarity of expression in response to wild-type (WT) Mtb. (**B, C**) Cluster 1 genes from A were subclustered based on the response to WT Mtb and the mutants. Uninfected, 12 hr, and 16 hr timepoints shown. (**A–C**) Blue-red gradient reflects relative expression within each row. (**D**) Volcano plots for the indicated conditions. *Tnf* and co-regulated gene *Nfkbiz* and subcluster 1B genes *Saa3, Marco, Lcn2* are indicated on each graph.

*Figure 1 continued on next page*

*Figure 1 continued*

RNAseq experiment performed once.

The online version of this article includes the following figure supplement(s) for figure 1:

**Figure supplement 1.** Wild-type and mutant Mtb strains are taken up into macrophages at similar rates.

*2003*); we thus considered the possibility that any attenuated mutant would elicit enhanced expression of subcluster 1B genes. To test this possibility, we obtained a well-characterized Mtb mutant. The deleted gene, *pckA*, catalyzes the first step in gluconeogenesis, and the mutant is highly attenuated for growth in macrophages (*Marrero et al., 2010*; *Figure 2—figure supplement 1B*). Infection with the *pckA* mutant did not elicit enhanced expression of subcluster 1B genes (*Figure 2—figure supplement 1C*), demonstrating that the enhanced response to PDIM or ESX-1 mutants is not a general response to mutants with impaired intracellular survival. These results suggest that PDIM and ESX-1 functions blunt induction of an inflammatory transcriptional program that includes multiple genes individually linked to control of tuberculosis (TB) infection.

## PDIM and ESX-1 blunt the later component of a biphasic TLR2-dependent transcriptional response to Mtb

Ingenuity pathway analysis of genes in subcluster 1B predicted MYD88 and NF-$\kappa$B as upstream regulators. To test these predictions and define upstream regulators, we profiled expression in BMDM from knockout mice. Consistent with pathway predictions, expression in response to either wild-type Mtb or the mutants was lost in *Myd88*-/- BMDM (*Figure 3A*). MYD88 functions as a signaling adapter for TLRs; we next sought to identify the relevant upstream TLR. Mtb produces multiple potential TLR2 antigens (*Brightbill et al., 1999*; *Gehring et al., 2004*; *Jung et al., 2006*; *Nair et al., 2009*; *Pathak et al., 2007*; *Pecora et al., 2006*; *Underhill et al., 1999*); we thus tested a role for TLR2 as the relevant upstream TLR. Similar to findings for MYD88, expression of representative genes upon infection with wild-type Mtb or PDIM or ESX-1 mutants was entirely lost in *Tlr2*-/- BMDM (*Figure 3A*). In contrast, *Tlr4*-/- knockout BMDM responded similarly to wild-type BMDM (*Figure 3B*). These results confirmed MYD88 and TLR2 as upstream regulators of macrophage transcriptional response component blunted by PDIM and ESX-1 function.

Given the preponderance of work using *Tnf* as a marker of TLR activation, we next examined the relationship between TLR2, *Tnf* expression, and PDIM and ESX-1 in the macrophage response to Mtb. *Tnf* did not cluster with 1B genes; instead *Tnf* was in cluster 3 (*Figure 1A*) together with genes minimally impacted by PDIM and ESX-1. Our RNAseq data additionally demonstrated that expression of *Tnf* peaked earlier post-infection than expression of subcluster 1B genes and then waned. At an MOI of 2:1, we found that induction of *Tnf* was very modest (less than twofold at the time of peak induction, *Figure 3C*), limiting our ability to reliably any decrease in *Tnf* expression. Infection at an MOI of 10:1 modestly enhanced expression of *Tnf* and delayed the time of peak expression (*Figure 3D*). At both MOI 2:1 and 10:1, we found that *Tnf* expression was very modestly impacted by PDIM and ESX-1 (*Figure 3C–D*). MOI of 5:1 gave similar kinetics and magnitude of *Tnf* expression to MOI of 10:1 (*Figure 3—figure supplement 1A*); we thus selected an MOI of 5:1 to minimize macrophage cell death while allowing us to reliably measure any impact of experimental interventions on *Tnf* expression. *Tnf* expression was partially lost in *Tlr2*-/- and *Myd88*-/- BMDM (*Figure 3E*), suggesting that additional PRRs likely contribute to *Tnf* expression in response to Mtb. We hypothesized that *Tnf* was part of a broader early TLR2-dependent transcriptional program; clustering genes across all infection conditions identified 17 genes with expression highly correlated with *Tnf* (*Figure 3—figure supplement 1B*). Testing expression of an additional representative gene from that cluster, *Nfkbiz*, confirmed a pattern of expression similar to *Tnf*. Similar to *Tnf*, expression was minimally impacted by PDIM and ESX-1 and was partially dependent upon MYD88 and TLR2 (*Figure 3C–E*). For Mtb infection of macrophages, *Tnf* transcription and release have been described as potentially dissociated (*Shi et al., 2005*). However, we found that similar to *Tnf* transcription, TNF release upon Mtb infection was partially dependent upon MYD88 and TLR2 (*Figure 3F*) and very modestly increased upon infection with PDIM- or ESX-1-mutant Mtb (*Figure 3G*). Our results thus suggest that Mtb infection of macrophages induces distinct early and late TLR2-dependent transcriptional responses, and that canonical

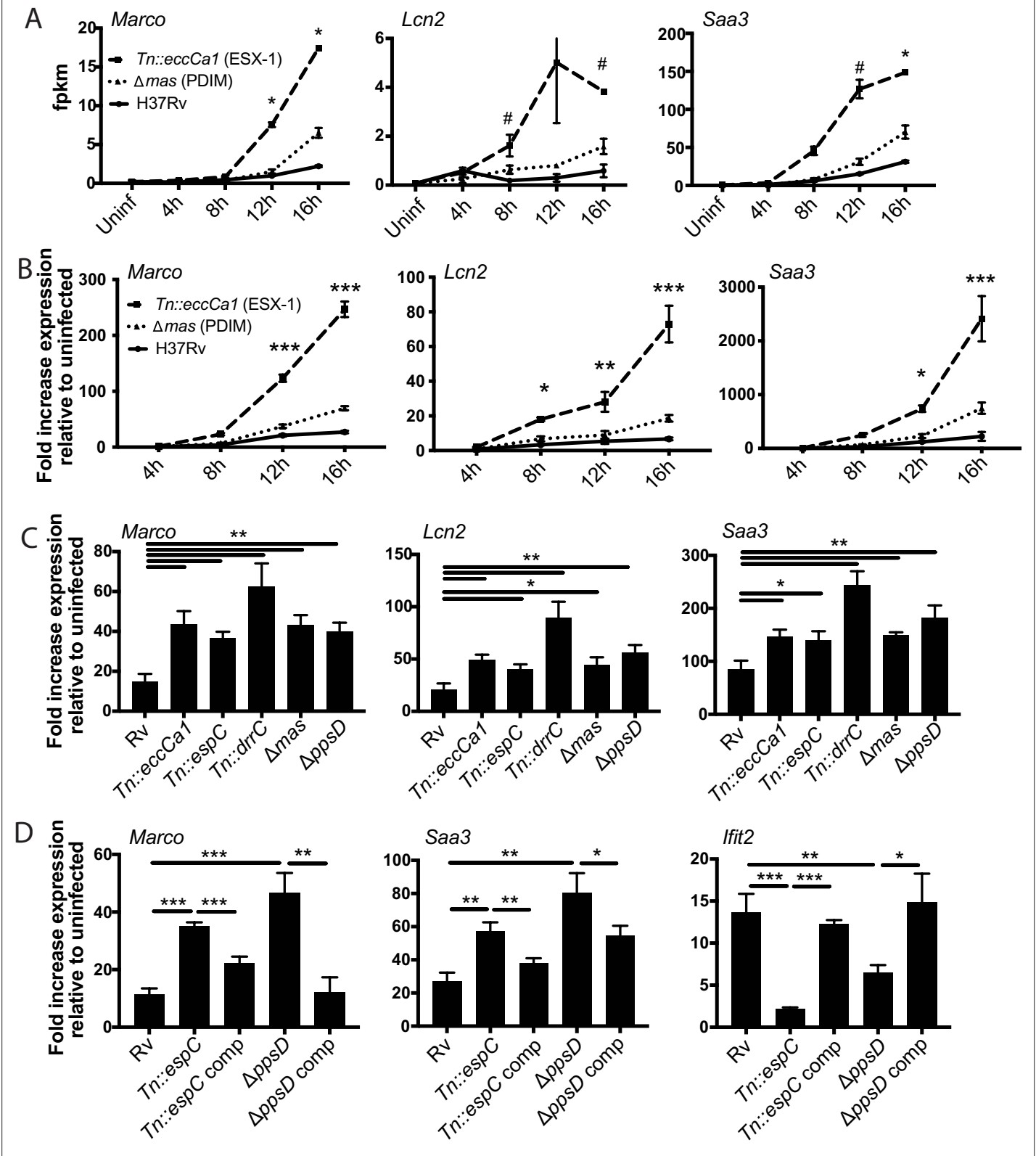

**Figure 2.** Infection with *Mycobacterium tuberculosis* (Mtb) phthiocerol dimycocerosate (PDIM) or ESX-1 mutants elicits enhanced expression of an inflammatory transcriptional program. (**A**) FPKM from RNAseq data (MOI 2:1) for representative genes from subcluster 1B. *p-value < 0.01 or #p-value < 0.05 for the comparison of PDIM (16 hr) or ESX-1 (12 hr) mutant-infected with Rv-infected, unpaired two-tailed t-test. (**B**) C57BL/6J bone marrow-derived macrophages (BMDM) were infected with the indicated strains at an MOI of 10:1. RNA was harvested at the indicated timepoints, and expression of the

*Figure 2 continued on next page*

*Figure 2 continued*
indicated genes was profiled by qPCR relative to GAPDH control. (**C–D**) C57BL/6J BMDM were infected with the indicated Mtb strains at an MOI of 2:1. RNA was harvested 24 hr post-infection, and expression of the indicated genes relative to GAPDH control was profiled using qPCR. (**B–D**) Mean ± SD of four replicates. *p-value < 0.01, **p-value < 0.001, ***p-value < 0.0001 unpaired two-tailed t-test. RNAseq experiment (**A**) performed once, (**B**) one of two independent experiments, (**C–D**) one of three independent experiments.

The online version of this article includes the following figure supplement(s) for figure 2:

**Figure supplement 1.** Enhanced expression is independent of BMDM mouse strain and specific to PDIM and ESX-1 mutants.

Mtb virulence factors that interfere with phagosomal membrane integrity preferentially blunt the later response.

## The observed biphasic transcriptional response is a fundamental feature of TLR2 signaling

We reasoned that the two-component TLR2 response we observed upon Mtb infection either could be pathogen-specific or could reflect a fundamental feature of TLR2 signaling. To distinguish between these possibilities, we treated BMDM with PAM3CSK4 or PAM2CSK4, synthetic agonists of TLR1/TLR2 and TLR2/TLR6, respectively, and profiled expression of genes representative of the early and late TLR2-dependent response to Mtb. Treatment with either synthetic ligand elicited expression of genes in the early component of the TLR2-dependent response to Mtb with a similar pattern of expression, peaking at 2–4 hr post-treatment before waning (*Figure 4A*, *Figure 4—figure supplement 1A*). PAM3CSK4 or PAM2CSK4 also elicited expression of genes in the later TLR2-dependent response component with delayed kinetics, evident by 4 hr post-infection but continuing to increase through 24 hr (*Figure 4B*, *Figure 4—figure supplement 1B*) As with Mtb infection, synthetic ligand-dependent expression of genes representative of both the early and late clusters was dependent upon TLR2 and the signaling adapter MYD88 (*Figure 4C–D*). PIM6 has been described to be the most potent TLR2 agonist in the Mtb PAMP repertoire (*Rodriguez et al., 2013*); we found that treating BMDM with purified PIM6 similarly induced the early and late response genes in a TLR2-dependent manner (*Figure 4E–F*). The two observed components of the TLR2-dependent response to Mtb infection thus appear to reflect inherent dynamics of TLR2 signaling in macrophages rather than dynamics specific to recognition of intact Mtb. None of the known adapters TRIF, TRAM, or TIRAP were required for induction of the late response (*Figure 4—figure supplement 1C–D*).

## The later component of the TLR2-dependent transcriptional response requires endosomal uptake

We next considered possible determinants of the two distinct TLR2 response components. We first considered that expression of genes in the later component may be driven by signaling through the TNF receptor initiated by the early component; however, late response component genes were expressed similarly in wild-type and TNF receptor knockout BMDM (*Figure 5—figure supplement 1A*). We then considered alternate hypotheses. PDIM and ESX-1 function preferentially undermine the second component of the response, and both interact with the phagosomal membrane. We thus hypothesized that the determinant of the distinct components may be spatial, with expression of the two sets of genes initiated at distinct subcellular sites. To distinguish between surface-initiated signal and endosome-specific signal, we used the dynamin inhibitor dynasore (*Macia et al., 2006*). Dynamin is required for the final step of formation of the endocytic vesicle, and in the context of LPS recognition by TLR4, dynamin has been used to dissect compartment-specific aspects of signaling (*Kagan et al., 2008*; *Rajaiah et al., 2015*; *Zanoni et al., 2011*). In the context of TLR2 on macrophages, dynamin does not change the surface and endosomal distribution of the receptor, but blocks uptake of PAM3CSK4 into the endosome (*Motoi et al., 2014*). We thus used dynasore and PAM3CSK4 to test whether uptake of synthetic TLR2 ligand into the endosome is required for activation of either component of the response. PAM3CSK4-induced expression of genes in the early cluster was not significantly changed by dynasore pre-treatment (*Figure 5A*). In contrast, PAM3CSK4-induced expression of genes in the late cluster was markedly reduced (*Figure 5B*).

We next wanted to test whether the compartment specificity of the two components of the TLR2-dependent pro-inflammatory response to synthetic ligand is similar for the response to Mtb. The

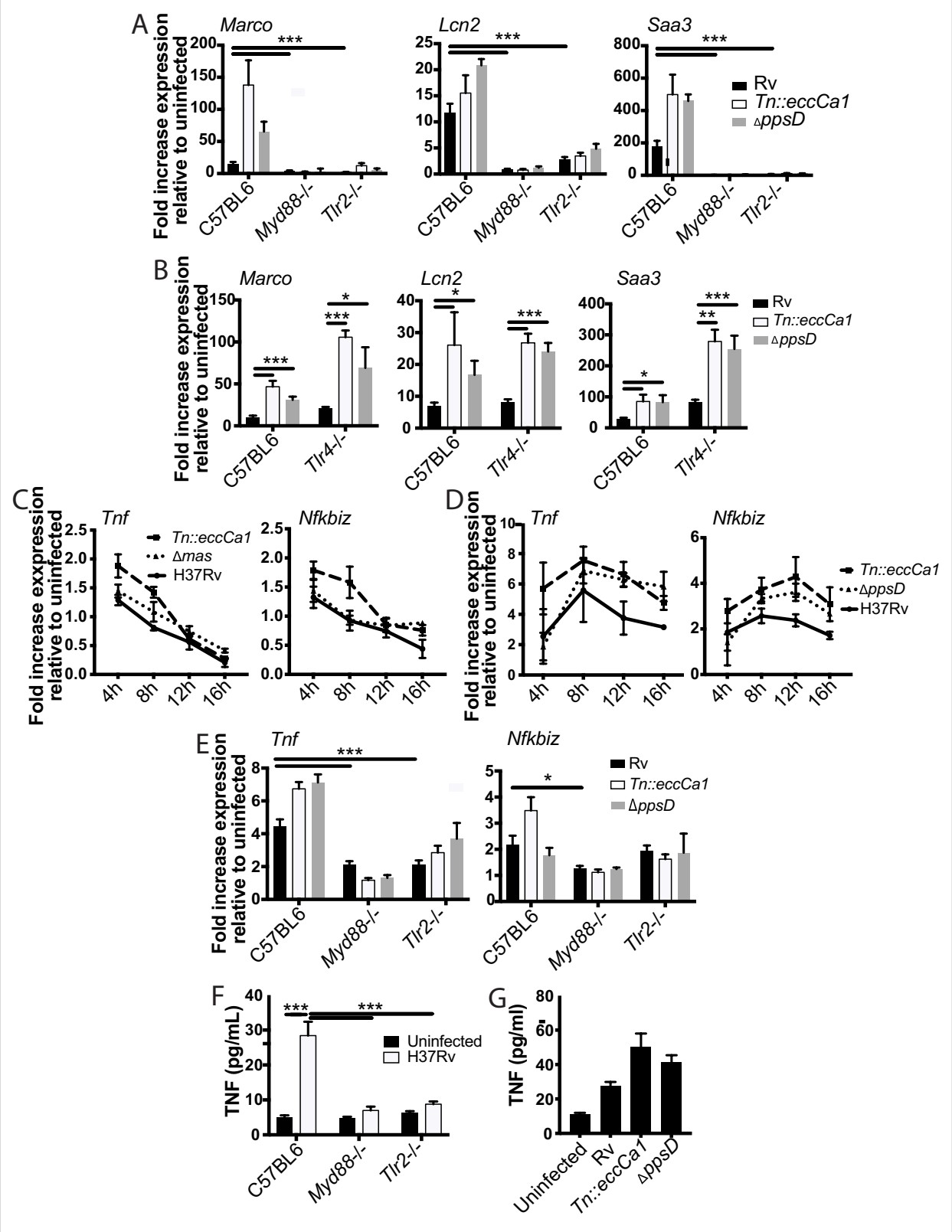

**Figure 3.** The identified two-component inflammatory response to *Mycobacterium tuberculosis* (Mtb) is dependent upon MYD88 and TLR2. (**A–B**) The indicated bone marrow-derived macrophages (BMDM) were infected with the indicated Mtb strains at an MOI of 2:1. RNA was harvested 24 hr post-infection. (**C–D**) C57BL/6J BMDM were infected with the indicated Mtb strains at an MOI of 2:1 (**C**) or 10:1 (**D**). RNA was harvested at the indicated timepoints post-infection. (**E**) The indicated BMDM were infected with the indicated Mtb strains at an MOI of 5:1. RNA was harvested 6 hr after infection.

*Figure 3 continued on next page*

*Figure 3 continued*

(**F–G**) The indicated (**F**) or C57BL/6J (**G**) BMDM were infected with the indicated Mtb strains at an MOI of 5:1. Supernatants were harvested 24 hr post-infection, and TNF was quantified by ELISA. Mean ± SD of four replicates. *p-value < 0.01, ***p-value < 0.0001 unpaired two-tailed t-test. (**A, C, E**) one of three independent experiments, (**B, D, F–G**) one of two independent experiments.

The online version of this article includes the following figure supplement(s) for figure 3:

**Figure supplement 1.** Expression of *Tnf* and co-regulated genes over time post-infection.

effect of dynasore on Mtb uptake has not previously been tested; using gentamicin protection assays, we found that dynasore significantly decreased Mtb uptake (*Figure 5—figure supplement 1B–C*). We then tested the effect of inhibiting Mtb uptake on expression of genes in the early and late clusters. Similar to treatment with synthetic ligand, inhibition of Mtb uptake with dynasore left expression of genes in the early TLR2-dependent cluster largely preserved (*Figure 5C*), but significantly blunted expression of genes in the late cluster (*Figure 5D*). Results were similar when BMDM were pre-treated

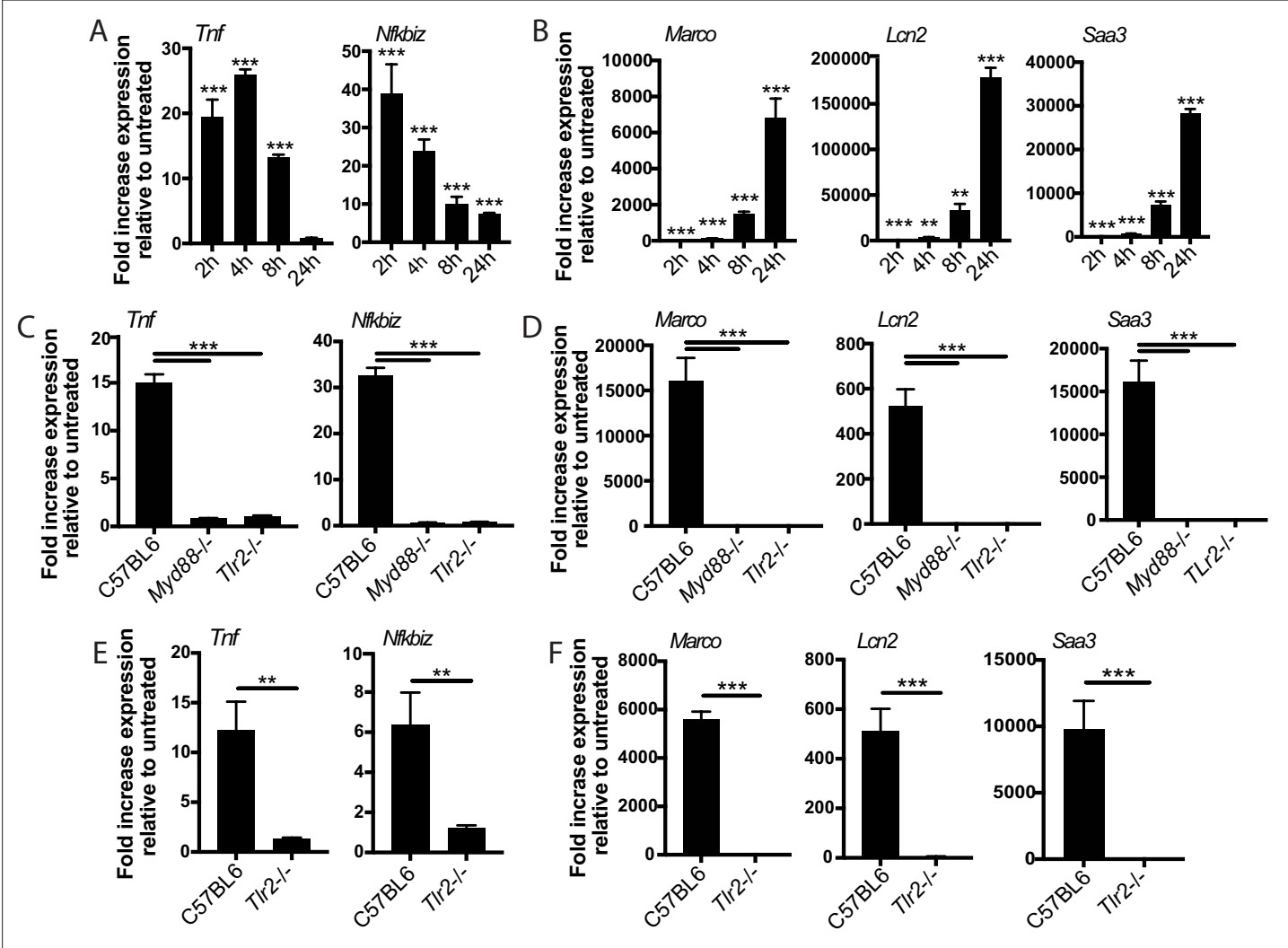

**Figure 4.** The two-component response is a fundamental feature of TLR2 activation. C57BL/6J (**A–B**) or the indicated (**C–F**) bone marrow-derived macrophages (BMDM) were treated with PAM3CSK 1 mu-g/ml (**A–D**) or PIM6 1 mu-g/ml (**E–F**) and RNA was harvested at the indicated timepoints (**A–B**), 2 hr (**C, E**), or 24 hr (**D, F**). qPCR was performed to quantitate expression of the indicated genes relative to GAPDH control. Mean ± SD for four replicates. **p-value < 0.001, ***p-value < 0.0001 unpaired two-tailed t-test. (**A–B**) One of two independent experiments, (**C–F**) one of three independent experiments.

The online version of this article includes the following figure supplement(s) for figure 4:

**Figure supplement 1.** PAM2CSK elicits the same two-component transcriptional response.

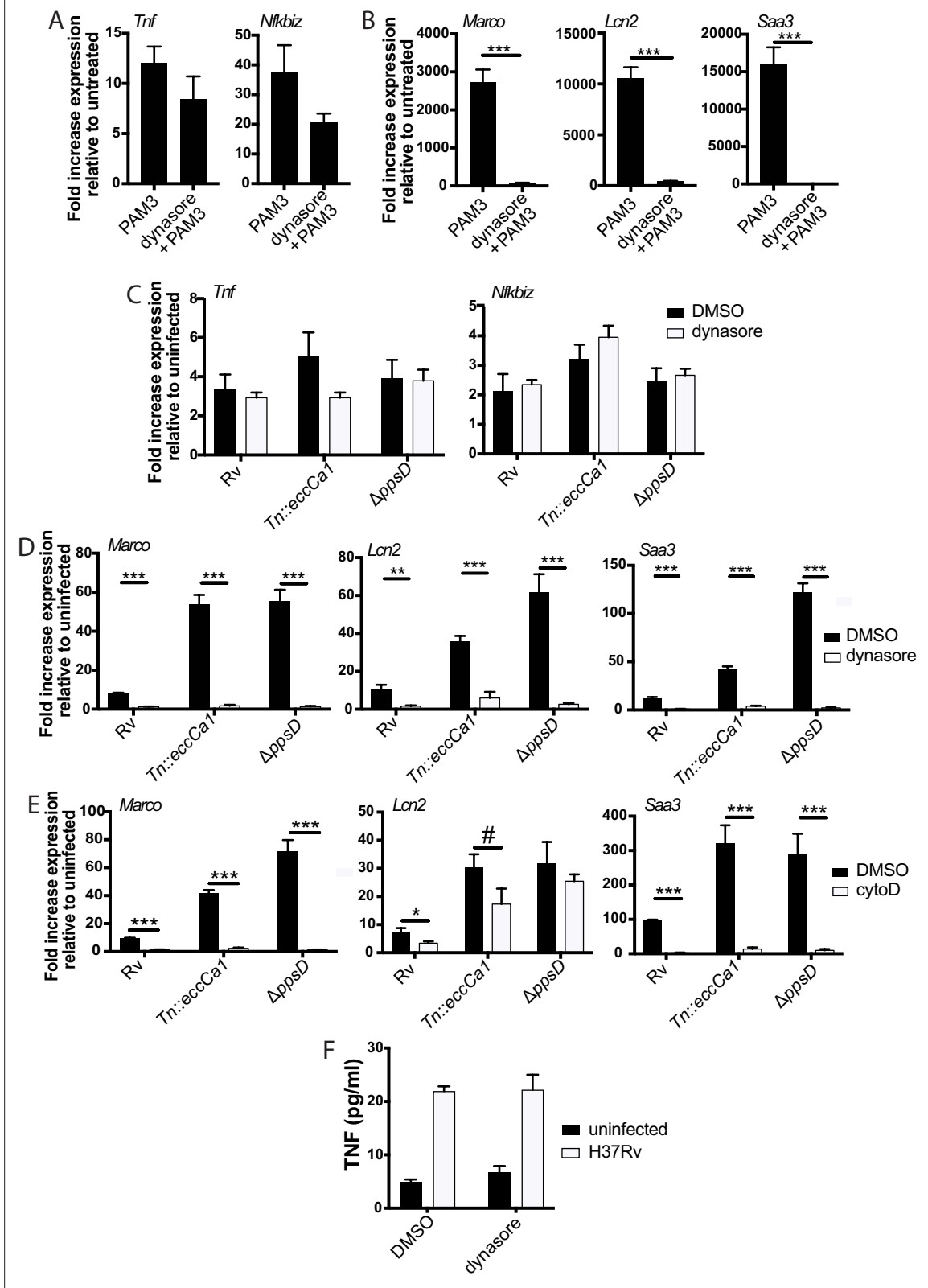

**Figure 5.** The later component of the TLR2-dependent transcriptional response requires endosomal uptake. Where indicated, C57BL/6J bone marrow-derived macrophages (BMDM) were pre-treated with dynasore 80 μM or cytochalasin D 10 μM. Cells were then treated with PAM3CSK4 1 μg/ml (**A–B**) or infected with the indicated *Mycobacterium tuberculosis* (Mtb) strains at an MOI of 5:1 (**C**) or 2:1 (**D–E**). RNA was harvested at 2 hr (**A**) 6 hr (**C**), or 24 hr (**B,D–E**). qPCR was performed to quantitate expression of the indicated genes relative to GAPDH control. (**E, F**) BMDM were infected with the indicated

*Figure 5 continued on next page*

*Figure 5 continued*

Mtb strains at an MOI of 5:1. Supernatants were harvested 24 hr post-infection, and TNF was quantified by ELISA. Mean ± SD for four replicates. #p-value < 0.05, *p-value < 0.01, ***p-value < 0.0001 unpaired two-tailed t-test. (**A–E**) One of three independent experiments, (**F**) one of two independent experiments.

The online version of this article includes the following figure supplement(s) for figure 5:

**Figure supplement 1.** The later component of the TLR2-dependent transcriptional response is independent of TNF signaling.

**Figure supplement 2.** The enhanced TLR2-dependent response to PDIM and ESX-1 mutants is independent of the type I IFN response.

with the actin polymerization inhibitor cytochalasin D (*Figure 5E*, *Figure 5—figure supplement 1C*), which has previously been used to distinguish innate immune signaling pathways initiated from the cell surface vs. the endosome (*Ip et al., 2010*; *Barbalat et al., 2009*; *Musilova et al., 2019*). Similar to the patterns observed for *Tnf* transcription, dynasore pre-treatment had minimal impact on TNF release (*Figure 5F*). These results suggest that while expression of early cluster genes can be initiated from the plasma membrane, expression of genes in the later cluster is dependent upon endosomal uptake.

Induction of type I IFNs in response to TLR2 activation has been previously linked to endosomal uptake (*Barbalat et al., 2009*; *Dietrich et al., 2010*; *Musilova et al., 2019*; *Stack et al., 2014*). We found that stimulation of BMDM with TLR2 agonists modestly induced type I IFNs (*Figure 5—figure supplement 2A, B*); this induction was dependent upon TLR2 (*Figure 5—figure supplement 2C*) and partially inhibited by dynasore pre-treatment (*Figure 5—figure supplement 2D*). However, the kinetics and magnitude of induction of *Ifnb1* were distinct from the late pro-inflammatory component of the TLR2 response, suggesting that the endosome-specific pro-inflammatory response is distinct from the type I IFN response. Consistent with established models, induction of the type I IFN response to Mtb was independent of TLR2 (*Figure 5—figure supplement 2E*). Our results suggest that while induction of the second component of the TLR2-dependent response is similar between Mtb and purified or synthetic TLR2 ligand, induction of type I IFNs is not part of this shared response.

## Full activation of the endosome-specific TLR2 response is dependent upon phagosome acidification

We next sought to understand how PDIM and ESX-1 function might undermine induction of the second component of the TLR2 response. Both PDIM and ESX-1 are required for induction of the type I IFN response to Mtb (*Barczak et al., 2017*; *Stanley et al., 2007*; *Figure 5—figure supplement 2F*), and interference between induction of type I IFNs and NF-$\kappa$B at the transcription factor level has previously been proposed in the macrophage response to other pathogens (*Scumpia et al., 2017*). We thus hypothesized that PDIM and ESX-1-facilitate induction of type I IFNs, and that type I IFN-activated transcription factors interfere with binding of NF-$\kappa$B-dependent transcription factors that contribute to the later component of the TLR2 response. To test this hypothesis, we profiled the macrophage response to Mtb in macrophages unable to mount a type I IFN response to infection. STING is strictly required for the type I IFN response to Mtb upstream of IRF3 activation (*Manzanillo et al., 2012*; *Figure 5—figure supplement 2E*). We predicted that if type I IFN-activated transcription factors blunt the TLR2 response, the response to wild-type Mtb would be increased in *Sting-/-* BMDM relative to wild-type BMDM. We additionally predicted that the response to wild-type Mtb and PDIM or ESX-1 knockouts would be equivalent in *Sting-/-* BMDM, as the type I IFN response would be similarly absent in response to all three Mtb strains. In fact, neither prediction tested correct (*Figure 5—figure supplement 2G*), suggesting that the mechanism through which PDIM and ESX-1 blunt the TLR2-dependent response to Mtb is independent of their role in type I IFN induction.

We then considered other ways that PDIM and ESX-1 function might interfere with the TLR2 response. Both PDIM and ESX-1 are required for phagosomal membrane damage (*Augenstreich et al., 2017*; *Manzanillo et al., 2012*; *Quigley et al., 2017*). Candida-mediated phagosomal membrane damage and sterile phagosomal membrane damage have both been described to interfere with phagosome acidification, potentially because of loss of the proton gradient across the membrane at sites of damage (*Eriksson et al., 2020*; *Westman et al., 2018*). We reasoned that PDIM- and ESX-1-mediated membrane damage might similarly contribute to the known limitation of acidification in Mtb-containing phagosomes. An ESX-1 mutant in *Mycobacterium marinum* has in fact

previously been shown to reside in a more highly acidified macrophage phagosome than wild-type *M. marinum* (**Tan et al., 2006**). Providing suggestive evidence for a link between phagosome acidification and the TLR2 response, inhibitors of phagosome acidification limit the MYD88-dependent response to *S. aureus* (**Ip et al., 2010**); this effect was attributed to a requirement for cathepsin activation within acidified lysosomes to process intact *S. aureus* and release TLR agonists. We thus hypothesized that PDIM and ESX-1 mediated membrane damage contributes to the limitation of phagosome acidification, and that limitation then impacts endosome-specific TLR2 activation.

To first test whether PDIM and ESX-1 function impact phagosome pH, we used the pH-sensitive fluorescent dye pHrodo. pHrodo labeling of Mtb has previously been used to quantify phagosomal pH around the mycobacterium (**Queval et al., 2017**). We labeled PDIM-mutant, ESX-1-mutant, or wild-type Mtb expressing GFP with pHrodo, then infected BMDM. CellProfiler (**Carpenter et al., 2006**) image analysis was used to identify GFP-Mtb; the corresponding pHrodo mean fluorescent intensity for each identified bacterium was then measured (**Figure 6—figure supplement 1A, B**). At 6, 12, or 24 hr post-infection, pHrodo mean fluorescent intensity around PDIM- or ESX-1-mutant Mtb was significantly higher than around wild-type Mtb (**Figure 6A**), suggesting that phagosomes containing PDIM- or ESX-1-mutant Mtb becomes relatively more acidic than phagosomes containing wild-type Mtb.

We then tested whether phagosome acidification enhances endosomal TLR2 signaling. We first confirmed that pre-treatment with concanamycin A, which inhibits the vacuolar ATPase, limits phagosome acidification in BMDM using both zymosan beads (**Figure 6—figure supplement 1C**) and infection with pHrodo-labeled PDIM and ESX-1 mutants (**Figure 6—figure supplement 1D**). We then pre-treated BMDM with concanamycin A prior to infection with Mtb. We found that while expression of the early cluster of genes was not significantly changed (**Figure 6B**), expression of the second cluster of genes was markedly diminished in the presence of concanamycin A (**Figure 6C**). These results suggested that phagosome acidification enhances the endosomal component of the TLR2-dependent response to Mtb. If acidification primarily drives the release of antigens from intact Mtb, as described for *S. aureus* (**Ip et al., 2010**), we would expect this acidification to be relevant upon infection with intact bacteria, but dispensable for the response to synthetic TLR2 ligand. Expression of genes in the early component of the response to synthetic TLR2 ligand was not changed by the addition of concanamycin A (**Figure 6D**). However, expression of genes in the later component of the response was markedly diminished by the addition of concanamycin A (**Figure 6E**). Similar results were obtained for the Mtb TLR2 ligand PIM6 (**Figure 6—figure supplement 1E–F**). These results suggest that the late component of TLR2 signaling is dependent on phagosome acidification entirely independent of the capacity to process pathogen and release TLR2 agonists. Taken together, our results are consistent with the hypothesis that Mtb-mediated damage of the phagosome membrane blunts the endosome-specific TLR2 response by limiting phagosome acidification. Further, the dependence on phagosome acidification suggests a potential mechanism through which compartment-specific TLR2 signaling is regulated. Signaling of the endosome-restricted TLRs, TLR7, and TLR9 is in fact strictly dependent upon endosomal acid-activated proteases (**Ewald et al., 2008**; **Park et al., 2008**), offering precedent for pH as a regulator of compartment-specific TLR signaling.

## PDIM and ESX-1 modulate TLR2-dependent infection outcomes in macrophages

We next sought to understand whether the interaction between PDIM/ESX-1 and TLR2 contributes to infection outcomes in macrophages. Mtb infection has been shown to drive macrophage cell death, including apoptosis (**Keane et al., 1997**), necrosis (**Danelishvili et al., 2003**), and ferroptosis (**Amaral et al., 2019**). PDIM has been described to specifically contribute to macrophage necrosis (**Quigley et al., 2017**); ESX-1 has also been shown to contribute to macrophage cell death after infection (**Augenstreich et al., 2017**; **Derrick and Morris, 2007**). In one study, pre-treatment of macrophages with a TLR4 or TLR2 agonist reduced Mtb-induced cell death (**Rojas et al., 1997**). Consistent with previous reports, we found that in wild-type macrophages PDIM and ESX-1 mutants induced less cell death than wild-type Mtb or complemented mutants (**Figure 7A–B**). We hypothesized that PDIM/ESX-1 interference with the late component of the TLR2 response might contribute to the cell death induced by wild-type Mtb; in that case, we would expect the enhanced macrophage survival observed upon infection with the PDIM or ESX-1 mutants to be lost or diminished in *Tlr2-/-* macrophages.

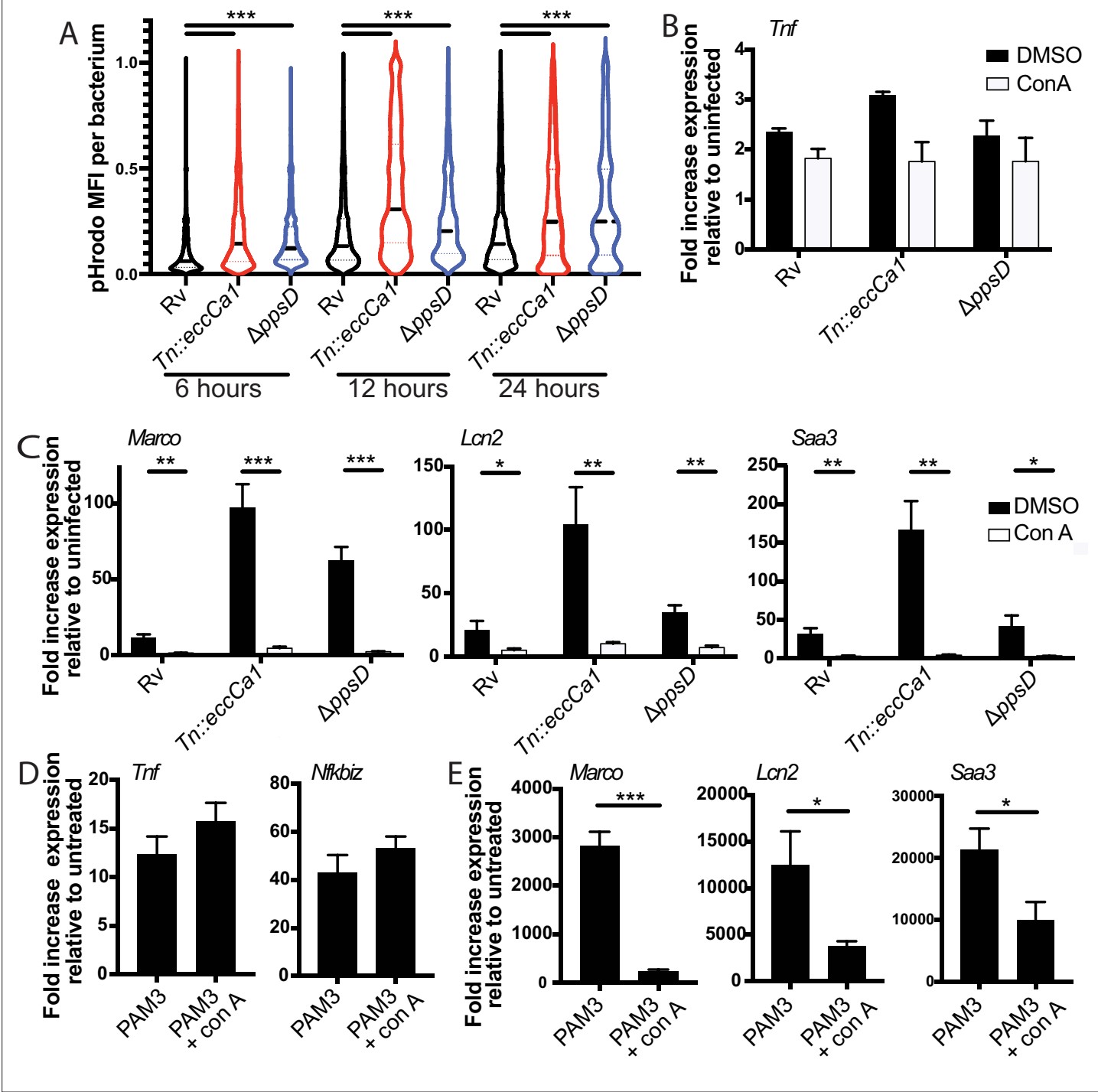

**Figure 6.** Full activation of the endosome-specific TLR2 response is dependent upon phagosome acidification. (**A**) The indicated *Mycobacterium tuberculosis* (Mtb) strains expressing GFP were labeled with pHrodo and used to infect C57BL/6J bone marrow-derived macrophages (BMDM) at an MOI of 3:1. After 4 hr, cells were washed to remove extracellular bacteria. Cells were fixed at 6, 12, and 24 hr post-infection and imaged. Bacteria were identified based on GFP signal, and pHrodo mean fluorescence intensity was measured around each bacterium. A minimum of 1703 bacteria were analyzed per group. (**B–C**) C57BL/6J BMDM were pre-treated with concanamycin A 50 μM, then infected with the indicated Mtb strains at an MOI of 5:1 (**B**) or 2:1 (**C**). (**C, D**) C57BL/6J BMDM were pre-treated with concanamycin A 50 μM, then stimulated with PAM3CSK4 1 mu-g/ml. RNA was harvested at 6 (**B**), 24 , (**C, E**) or 2 hr (**D**) post-infection. qPCR was performed to quantitate expression of the indicated genes relative to GAPDH control. Mean ± SD for four replicates. #p-value < 0.03, *p-value < 0.01, **p-value < 0.001, ***p-value < 0.0001, unpaired two-tailed t-test. (**A–E**) One of three independent experiments.

*Figure 6 continued on next page*

*Figure 6 continued*

The online version of this article includes the following figure supplement(s) for figure 6:

**Figure supplement 1.** Concanamycin A blocks phagosome acidification.

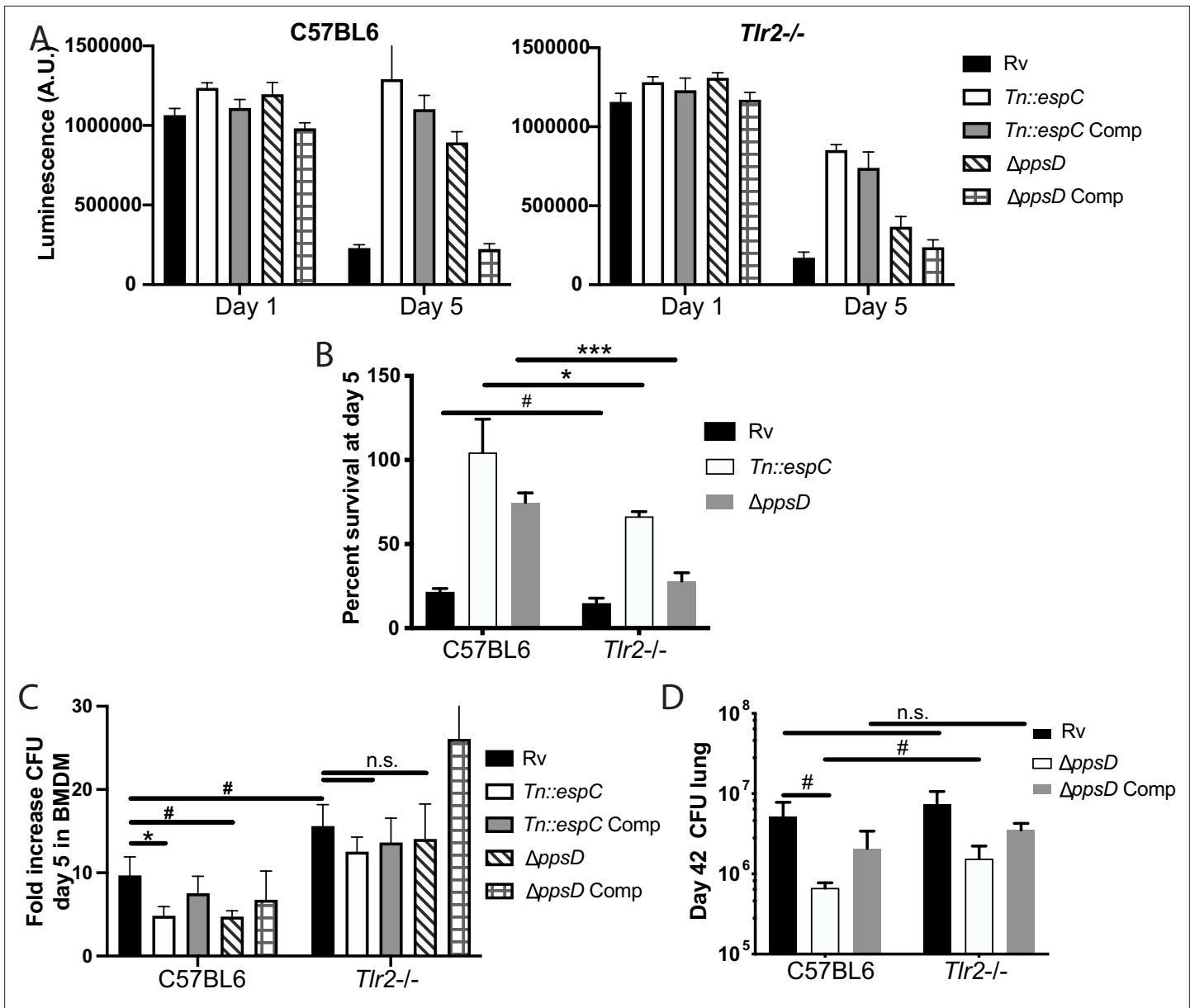

**Figure 7.** Phthiocerol dimycocerosate (PDIM) and ESX modulate TLR2-dependent infection outcomes in macrophages and mice. (**A–C**) The indicated bone marrow-derived macrophages (BMDM) were infected with the indicated *Mycobacterium tuberculosis* (Mtb) strains at an MOI of 5:1 (**A–B**) or 2:1 (**C**). (**A–B**) Cell survival was determined using a CellTiterGlo luminescence assay at the indicated days post-infection. (**C**) At day 5 post-infection, cells were washed, lysed, and plated for CFU. (**A–C**) Mean ± SD for four replicates. #p-value < 0.05, *p-value < 0.01 unpaired two-tailed t-test. (**D**) C57BL/6J or *Tlr2-/-* mice were infected with ~200 cfu of the indicated Mtb strains; 42 days post-infection, mice were euthanized and lungs were harvested and plated in serial dilutions to determine CFU Mean ± SD for five mice per condition (one C57BL6/ppsD plate discarded for mold contamination – four replicates for that condition). #p-value < 0.05, unpaired two-tailed t-test. (**A–C**) One of three independent experiments, (**D**) one of two independent experiments.

The online version of this article includes the following figure supplement(s) for figure 7:

**Figure supplement 1.** The indicated mouse strains were infected with the indicated *Mycobacterium tuberculosis* (Mtb) strains via low-dose aerosol infection.

Infection with wild-type Mtb induced a similar degree of cell death in wild-type and *Tlr2-/-* macrophages. However, the resistance to cell death observed in wild-type macrophages infected with PDIM or ESX-1 mutants was partially lost in *Tlr2-/-* macrophages (*Figure 7A–B*). These results suggest that PDIM and ESX-1 interference with TLR2-dependent responses contributes to macrophage cell death following infection.

We next sought to test whether the interaction between PDIM/ESX-1 and TLR2 impacts Mtb survival and growth in macrophages. PDIM and ESX-1 mutants have an attenuated growth phenotype in macrophages (*Camacho et al., 1999*; *Stanley et al., 2003*). We hypothesized that if TLR2-dependent responses contribute to this growth restriction, PDIM and ESX-1 mutants should grow more robustly in *Tlr2-/-* BMDM than in wild-type BMDM. Alternatively, if PDIM- and ESX-1-mutant growth restriction is entirely independent of TLR2-dependent responses, those mutants should grow similarly in wild-type and *Tlr2-/-* BMDM. Using a low MOI to minimize induction of macrophage cell death, we infected wild-type or *Tlr2-/-* BMDM with our wild-type or mutant Mtb strains. As expected, PDIM and ESX-1 mutants grew less well than wild-type Mtb in C57BL/6J BMDM (*Figure 7C*). Growth of wild-type Mtb was modestly enhanced in *Tlr2-/-* macrophages relative to wild-type macrophages; further, the PDIM and ESX-1 mutants grew significantly more robustly in the *Tlr2-/-* BMDM than in wild-type BMDM, with growth similar to wild-type Mtb (*Figure 7C*). These results suggest that TLR2-dependent responses contribute to growth restriction of the PDIM and ESX-1 mutants in macrophages. Together with our macrophage survival data, these results indicate that PDIM and ESX-1-mediated interference with TLR2-dependent responses contributes to the pathogenesis of Mtb infection in macrophages.

## PDIM modulates TLR2-dependent infection outcomes in mice

Several studies have assessed the role of TLR2 in infection outcomes in mice. Some studies have shown a modest increase in Mtb growth in *Tlr2-/-* mice relative to wild-type mice (*Drennan et al., 2004*; *McBride et al., 2011*; *McBride et al., 2013*), while others have shown no difference in CFU. Most studies have shown increased lung pathology in *Tlr2-/-* mice, with larger infiltrates, less organization, and more inflammatory cells described as key features in various studies (*Drennan et al., 2004*; *McBride et al., 2013*; *Bafica et al., 2005*). *Tlr2-/-* mice have been shown to have increased Mtb dissemination to liver and spleen (*Drennan et al., 2004*) and a more rapid progression to death following infection (*Drennan et al., 2004*; *Bafica et al., 2005*; *Reiling et al., 2002*). In aggregate, these data suggest that TLR2 plays a somewhat modest role in host control of TB infection. We hypothesized that PDIM and ESX-1 interference with the late TLR2-dependent response limits the contribution TLR2 makes to host control of Mtb infection. To test this hypothesis, we compared infection outcomes in wild-type and *Tlr2-/-* mice. As previous profiling had demonstrated that CFU and pathology diverge by 6 weeks post-infection (*McBride et al., 2013*), we selected this timepoint for study. A limited number of mutant mice could be obtained for these studies; we thus focused on the interaction between PDIM and TLR2.

C57BL/6J and *Tlr2-/-* mice were infected with wild-type, PDIM-mutant, and complemented PDIM-mutant Mtb (*Figure 7—figure supplement 1A*). At the 6 week timepoint, as expected, PDIM-mutant Mtb growth was restricted relative to wild-type Mtb growth in C57BL/6J mice. Wild-type Mtb grew similarly in the lungs of C57BL/6J and *Tlr2-/-* mice. In contrast, PDIM-mutant Mtb had increased growth in the lungs of *Tlr2-/-* mice relative to C57BL/6J mice (*Figure 7D*), suggesting that PDIM-mediated interference with activation of components of the TLR2-dependent response contributes to the capacity of the bacterium to grow in lung. Histopathologically, the lungs of C57BL/6J mice infected with wild-type Mtb demonstrated defined areas of inflammation by 6 weeks post-infection (*Figure 7—figure supplement 1B*), with dense inflammatory infiltrates composed of foamy and non-foamy macrophages and lymphocytes clusters (*Figure 7—figure supplement 1C*). Lungs of C57BL/6J mice infected with PDIM-mutant Mtb showed trends toward fewer areas of inflammation and smaller lesion sizes (*Figure 7—figure supplement 1B*). Examination of the regions of cellular infiltration in mice infected with PDIM-mutant Mtb were notable for similar presence of foamy macrophages and lymphocytes (*Figure 6—figure supplement 1B*). In *Tlr2-/-* mutant mice infected with PDIM-mutant Mtb, infiltrates showed a trend toward more numerous and larger areas of involvement than was observed in C57BL/6J mice (*Figure 7—figure supplement 1B–C*). In total, our data suggest that PDIM modulation of TLR2-dependent responses contributes to pathogenesis in vivo. Growth of the

PDIM mutant is only partially restored in *Tlr2-/-* mice, indicating that additional mechanisms contribute to the attenuation of PDIM mutants in vivo.

## Discussion

Accumulating data suggest that pathogenic bacteria evolve strategies for evading the components of immunity most critical for controlling their survival and replication. Multiple intracellular pathogens damage the phagosomal membrane in the course of pathogenesis, raising the question of whether this shared function reflects a convergent evolutionary strategy. While phagosomal membrane damage has been proposed to benefit the bacterium in the host-pathogen standoff, the mechanisms through which that benefit might accrue have not been well established experimentally. In the case of pathogens recognized by TLR2, our results raise the possibility that damaging the phagosomal membrane may serve as a common strategy to limit effective inflammation.

Previous work has suggested that TLR2 can signal from the plasma membrane or endosome. Investigation of the mechanisms and consequences of TLR2 signaling have primarily focused on pathogen-specific induction of type I IFNs (*Barbalat et al., 2009*; *Dietrich et al., 2010*; *Musilova et al., 2019*; *Stack et al., 2014*) and TNF (*Ip et al., 2010*; *Brandt et al., 2013*), largely in response to *S. aureus* exposure or viral infection. Our results support a model in which TLR2 activation in fact drives distinct compartment-specific pro-inflammatory transcriptional responses, reflected in both the sets of genes expressed and the kinetics of induction. Although a fundamental feature of TLR2 signaling, the two transcriptional response components are likely to have different relevance in the context of individual infections. In the case of TB, TNF, a component of the early response, is known to be critical for infection control. However, the later response component includes expression of multiple genes demonstrated to be important for both cell intrinsic control of Mtb and priming of the adaptive immune response. Our results demonstrating an interaction between TLR2 and PDIM or ESX-1 for infection outcome suggest that undermining this second component contributes to Mtb's success as a pathogen. More broadly, our results suggest that expanding beyond TNF and type I IFNs as markers of TLR activation may offer both new insights into mechanisms and consequences of TLR signaling and into the links between TLR activation and control of pathogenic infection.

Our work suggests one potential mechanism through which PDIM and ESX-1 contribute to the pathogenicity of Mtb. PDIM has previously been shown to interfere with an effective MYD88 inflammatory response, as measured by the outcomes of macrophage recruitment to *M. marinum*-containing lesions in vivo and iNOS production (*Cambier et al., 2014*). In that work, this effect was hypothesized to be attributable to the unmasking of TLR agonists on the mycobacterial surface in the absence of PDIM, an abundant outer membrane lipid. Our results confirm an effect of PDIM on MYD88-dependent inflammation but point toward a different potential molecular interaction between PDIM and inflammation – namely that PDIM limits an endosomal component of the TLR2 response by limiting phagosome acidification. Two lines of evidence support the latter proposed mechanism. First, the effect we observe on TLR2-dependent inflammation is shared between PDIM and ESX-1. ESX-1-mediated secretion is not known to be required for localization of any known TLR2 agonist and in fact Mtb has multiple distinct TLR2 ligands; thus the common effect of PDIM and ESX-1 on TLR2-dependent inflammation is unlikely to be due to masking of TLR2 agonists. Second, PDIM and ESX-1 only minimally impact expression of the early TLR2-dependent gene cluster, suggesting that the inherent capacity of TLR2 to 'recognize' cognate ligand on the bacterium is similar between wild-type Mtb and ESX-1- or PDIM-mutant Mtb.

Together with previous work identifying Mtb factors that interfere with TLR2 activation, our work points toward an explanation for the puzzling disparity between the number of identified TLR2 ligands that Mtb possesses and the relatively modest phenotype of Mtb infection in TLR2 knockout mice. Work from other groups has identified mycobacterial strategies for interfering with TLR2 activation, primarily studied through an impact on TNF expression and release. The secreted hydrolase Hip1 has been shown to blunt the secretion of cytokines, including TNF, following infection (*Madan-Lala et al., 2011*). Recently, the surface lipid sulfolipid-1 was shown to interfere with surface recognition of TLR2 agonists (*Blanc et al., 2017*). Our work suggests that the canonical Mtb virulence factors PDIM and ESX-1 function to blunt a distinct, endosome-specific component of the TLR2 response, and that this interference in fact modulates infection outcomes in macrophages and in mice. In aggregate, these results suggest that mycobacteria have evolved multiple strategies to undermine TLR2 activation from

both the surface and endosome. Adding to the complexity of TLR2-dependent phenotypes in TB infection, studies of the relationship between Mtb TLR2 agonists and IFN-dependent functions have shown that TLR2 activation can dampen IFN-dependent gene expression and cell functions (*Pecora et al., 2006*; *Banaiee et al., 2006*). Ultimately developing new strategies for treating TB will rely on a deep understanding of the pathogenesis of infection that enables the rational selection of therapeutic targets. Host-directed therapies enhancing the host-protective components of TLR2 activation might offer a path to a more effective inflammatory response to Mtb and ultimately more effective sterilization of TB infection.

## Materials and methods
### Bacterial strains and culture
The indicated *Mtb* strains were grown in Middlebrook 7H9 broth (Difco) with Middlebrook OADC (BD), 0.2% glycerol, and 0.05% Tween-80. Mtb strains H37Rv, H37Rv*Tn::eccCa1*, H37Rv*ppsD*, H37RvΔ*pps-D*::pMV261:: ppsD, and H37RvΔ*mas* were characterized in *Barczak et al., 2017*. H37Rv*Tn::espC* was grown from a published transposon library in H37Rv (*Barczak et al., 2017*). The complement was generated by cloning the *espACD* operon from H37Rv into the Kpn and XbaI sites of shuttle plasmid pMV261 (*Stover et al., 1991*) (F primer atgacagatcggcctagctagg R primer attgtgagcccagtcgggaaa).

### Macrophage infections
BMDM were isolated and differentiated in DMEM containing 20% FBS (Cytiva) and 25 ng/ml rm-M-CSF (R&D Systems) as previously described (*Barczak et al., 2017*). Infections were carried out as previously described (*Barczak et al., 2017*; *Stanley et al., 2014*). Briefly, Mtb strains used were grown to mid-log phase, washed in PBS, resuspended in PBS, and subjected to a low-speed spin to pellet clumps. BMDM were infected at the indicated MOI, allowing 3–4 hr for phagocytosis. Cells were then washed once with PBS, and media was added back to washed, infected cells. The MOI used for each timepoint was selected to maximize signal while minimizing infection-associated cell death.

### Mouse strains
C57BL/6J (Jackson Laboratories strain #000664), BALB/c (Jackson Laboratories strain #000651), *Sting-/-* (C57BL/6J-*Sting*$^{gt}$/J, Jackson Laboratories strain # 017537), *Tlr4-/-* (Jackson Laboratories strain #007227) *Tlr2-/-* (B6.129-*tlr2*$^{tm1Kir}$/J, Jackson Laboratories strain #004650), *Myd88-/-* (B6.129P2(S-JL)-*Myd88*$^{tm1.1Defr}$/J, Jackson Laboratories strain #009088), *TNFAR-/-* (B6.129S-*Tnfrsf1a*$^{tm1lmx}$ *Tnfrsf1b*$^{tm1lmx}$/J, Jackson Laboratories strain #003243), and *Trif-/-* (C57BL/6j-*Ticam1*$^{Lps2}$/J, Jackson Laboratories strain #005037) mice were used for the preparation of BMDM.

### RNA isolation and qPCR
Infected BMDM were lysed at designated timepoints following infection with β-ME-supplemented Buffer RLT (Qiagen). RNA was isolated from lysate using an RNEasy kit (Qiagen) supplemented with RNase-free DNase I digest (Qiagen), both according to manufacturer's protocol. cDNA was prepared using SuperScript III (Thermo Fisher Scientific) according to manufacturer's protocol. qPCR was performed using PowerUP SYBR Green (Thermo Fisher Scientific) and primers specific to investigated genes relative to GAPDH control.

### RNAseq
Poly(A) containing mRNA was isolated from 1 µg total RNA using NEBNext Poly(A) mRNA Magnetic Isolation Module (New England Biolabs). cDNA libraries were constructed using NEBNext Ultra II Directional RNA Library Prep Kit for Illumina and NEBNext Multiplex Oligos for Illumina, Index Primers Sets 3 and 4 (New England Biolabs). Libraries were sequenced on an Ilumina NextSeq500. Bioinformatic analysis was performed using the open source software GenePattern (*Drennan et al., 2004*; *Reiling et al., 2002*). Raw reads were aligned to mouse genome using TopHat, and Cufflinks was used to estimate the transcript abundance. FPKM values obtained by Cufflinks were used to plot heatmaps (*Zenkova et al., 2019*). Three biological replicates for each condition were performed; replicates that failed QC metrics were not included in heatmaps and clustering. K-means clustering was performed in R and functional analysis was performed using IPA (*Krämer et al., 2014*) (QIAGEN Inc, https://www.

qiagenbio-informatics.com/products/ingenuity-pathway-analysis/). RNAseq data is accessible on the NCBI GEO website GSE144330.

## TNF ELISAs

BMDM were infected at an MOI of 5:1 as described above. Following a 4 hr phagocytosis, BMDM were washed, and BMDM media was added back. At 24 hr post-infection, supernatants were collected for quantitation of TNF-α using an ELISA Ready-SET-Go! kit according to the manufacturer's protocols. (Thermo Fisher Scientific). Four replicates were performed per condition based on the determination that this would give 80% power to detect a 20% difference between samples.

## Imaging of pHrodo-labeled bacteria in macrophages

Imaging of pHrodo-labeled Mtb as described in *Queval et al., 2017*. Mtb strains were grown to mid-log phase, then washed twice with an equal volume of PBS. Mtb was then resuspended in 100 mM $NaHCO_3$ with 0.5 M pHrodo dye (Invitrogen) and incubated at room temperature in the dark for 1 hr. The labeled cells were then washed three times with PBS, after which BMDM infections were performed as described above. At the indicated time post-infection, infected BMDM were washed with PBS and fixed in 4% paraformaldehyde. Nuclei were labeled with DAPI (1.25 mu-g/ml). Cells were then imaged on a Zeiss Elyra microscope with a 40× oil objective or a TissueFAXS confocal microscope with a 40× objective. Images were imported into CellProfiler (*Carpenter et al., 2006*) for analysis. Bacterial outlines were identified based on GFP signal; the outline was then expanded by five pixels, and pHrodo fluorescence intensity within the expanded outline was determined.

## CFU quantitation

Bacteria were prepared as described above, and added to BMDM at an MOI of 2:1. After 4 hr, cells were lysed in 0.5% Triton X-100, diluted in 7H9 media, and plated on 7H10 plates for colony enumeration. For gentamicin killing assay, cells were treated with dynasore (80 µM) prior to infection where indicated. Following the 4 hr phagocytosis, cells were washed in PBS with gentamicin (32 mu-g/ml), then resuspended in bone marrow macrophage media with gentamicin (32 mu-g/ml). After allowing 2 hr for killing of extracellular bacteria, cells were washed, lysed, diluted, and plated for colony enumeration.

## Quantitation of MFI for pHrodo-labeled zymosan beads

C57BL/6J BMDM were plated in an eight-chamber slide. Cells were pre-treated with concanamycin A (50 µM) or DMSO carrier for 15 min. Media was then removed, and pHrodo red zymosan bioparticles (Invitrogen) were added at 0.5 mg/ml in BMDM media with concanamycin A or DMSO carrier. After 2 hr, media was removed and cells were washed once with PBS. Cells were then fixed in 4% paraformaldehyde and stained with DAPI (1.25 mu-g/ml). Cells were imaged on a Zeiss Elyra PS.1 microscope with a 20× objective. Images were analyzed using a CellProfiler image analysis pipeline. DAPI-stained nuclei were identified and counted and integrated red pHrodo fluorescence was measured for each image.

## Macrophage survival assays

C57BL/6J or TLR2-/- BMDM were plated in 96-well format and infected with Mtb strains at an MOI of 5:1. Cells were harvested day 1 and day 5 post-infection using a CellTiter-Glo Luminescent Cell Viability Assay Kit (Promega) in accordance with the manufacturer's instructions. After gentle agitation and 10 min incubation, wells were read on a Tecan Spark 10 M luminescent plate reader. Media was replenished every 48 hr post-infection.

## Mouse infections

C57BL/6J (Jackson Laboratories strain #000664) or *Tlr2-/-* (Jackson Laboratories strain #021302) mice were infected via low-dose aerosol exposure with an AeroMP (Biaera Technologies). Three to five mice per condition were harvested at day 0 to quantify inoculum. Six weeks post-infection, mice were euthanized in accordance with AALAC guidelines, and lungs were harvested for CFU and histopathology. Formalin-fixed lungs were embedded in paraffin, sectioned, and stained with hematoxylin

and eosin by the MGH histopathology core. Images were acquired on a TissueFAXS slide scanner (TissueGnostics).

## Acknowledgements

The authors would like to thank Drs Roi Avraham, Bryan Bryson, Sarah Fortune, and Jonathan Kagan for critical manuscript review and Dr Lenette Lu and the laboratories of Drs Marcia Goldberg and Cammie Lesser for helpful discussions. We would additionally like to thank Dr Sabine Ehrt for the pckA mutant, parent, and complement strains. The work was funded in part by an MGH Transformative Scholar Award (AKB) and made possible by help from the Harvard University Center for AIDS Research (CFAR), an NIH funded program (P30 AI060354).

## Additional information

### Funding

| Funder | Grant reference number | Author |
| --- | --- | --- |
| MGH Transformative Scholar Award | | Amy K Barczak |

The funders had no role in study design, data collection and interpretation, or the decision to submit the work for publication.

### Author contributions

Amelia E Hinman, Conceptualization, Data curation, Formal analysis, Investigation, Methodology, Writing – review and editing; Charul Jani, Conceptualization, Formal analysis, Visualization, Writing – review and editing; Stephanie C Pringle, Wei R Zhang, Formal analysis, Investigation, Writing – review and editing; Neharika Jain, Formal analysis, Writing – review and editing; Amanda J Martinot, Formal analysis, Supervision, Writing – review and editing; Amy K Barczak, Conceptualization, Data curation, Formal analysis, Funding acquisition, Investigation, Methodology, Supervision, Writing – original draft, Writing – review and editing

### Author ORCIDs

Amy K Barczak http://orcid.org/0000-0003-3806-2381

### Ethics

This study was performed in accordance with guidelines of the Massachusetts General Hospital Institutional Care and Use Committee, under the approved protocols 2014N000297 and 2014N000311.

### Decision letter and Author response

Decision letter https://doi.org/10.7554/eLife.73984.sa1
Author response https://doi.org/10.7554/eLife.73984.sa2

## Additional files

### Supplementary files

• Supplementary file 1. Table of FPKM values for RNAseq data used to generate the heatmap for *Figure 1*.

• Transparent reporting form

### Data availability

RNAseq data is accessible on the NCBI GEO website GSE144330.

The following dataset was generated:

| Author(s) | Year | Dataset title | Dataset URL | Database and Identifier |
|-----------|------|---------------|-------------|-------------------------|
| Barczak AK | 2020 | RNAseq data for murine BMDM infected with wild-type Mycobacterium tuberculosis, a PDIM mutant, or an ESX-1 mutant | https://www.ncbi.nlm.nih.gov/geo/query/acc.cgi?acc=GSE144330 | NCBI Gene Expression Omnibus, GSE144330 |

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
