## [Editor Report]

This article provides insight into the kinetics of TLR2 mediated immune responses and the roles for Esx1 and PDIM in these responses. These results demonstrate two distinct components of the TLR2 response and show the relevance of these responses to *Mycobacterium tuberculosis* pathogenesis. These findings are important and of interest to the broader field of host cell interactions with intracellular pathogens.

---

## [Decision Letter]

[Editors’ note: the authors submitted for reconsideration following the decision after peer review. What follows is the decision letter after the first round of review.]

Thank you for submitting your work entitled "*Mycobacterium tuberculosis* canonical virulence factors interfere with compartment-specific TLR2 signaling" for consideration by *eLife*. Your article has been reviewed by 3 peer reviewers, one of whom is a member of our Board of Reviewing Editors, and the evaluation has been overseen by a Senior Editor. The following individual involved in review of your submission has agreed to reveal their identity: Ousman Tamgue (Reviewer #3).

Our decision has been reached after consultation between the reviewers. Based on these discussions and the individual reviews below, we regret to inform you that your work will not be considered further for publication in *eLife*.

The reviewers all agreed this was a very interesting study, but after much discussion decided that the manuscript requires further experiments that realistically would take more than 2 months to complete. In this case, it is *eLife* policy to reject the paper, encouraging the authors to resubmit a revised version that includes the experiments requested by the reviewers. The revised version in this case would be considered as a new submission. However, we understand that this is a very difficult time to address requests for experiments given the widespread laboratory shutdowns in response to COVID-19.

*Reviewer #1:*

The article entitled "*Mycobacterium tuberculosis* canonical virulence factors interfere with compartment-specific TLR-2 signaling" by Hinman et al. provides insight into the TRL2 mediated two-component immune response acting through two clusters of genes at an early and late stage of infection in vitro. The authors focus on the late stage TLR2 signaling cascade, which they show is dependent on vesicle acidification. In general, the experiments were rigorously performed and well-presented.

A major concern is the variation in MOI that was used throughout the manuscript. The authors mention that "the MOI used for each time point was selected to maximize signal while minimizing infection-associated cell death", but how do the authors correct for different MOIs across experiments? For example, in Figure-1, MOI 2 of infection was used for RNAseq whereas MOI 10 was used for qPCR analysis. The authors should mention the reason behind switching from a lower dose infection (MOI 2) to a higher dose infection (MOI 10). It would be interesting to see an MOI dependent expression pattern of Saa3, Lcn2, and Marco at 16h post-infection in figure 1. As another example, TNF expression was studied at different time points with MOI 10 in figure 2C which suggested that at 8hr post-infection, the expression of TNF reaches its peak. But then the authors reduced the MOI to 5 and time point to 6hr post-infection for the TNF expression analysis in Figure 3D.

For Figure 4, the authors could show a time-dependent pH measurement for each Mtb strain (eg. 12h, 24h, 48h) to more precisely coordinate the timing of acidification with the second wave of TLR signaling. It would be nice if the authors provide a representative 5 pixel expanded microscopy image as they mentioned in Material and Methods in the supplementary data for the phagosomal pH analysis of each Mtb strain.

*Reviewer #2:*

The ESX-1 type VII secretion system is critical for virulence of *M. tuberculosis* and has been proposed to function in part by perforation of phagosomal membrane. In a previous study, the corresponding author found that the complex lipid PDIM is linked to ESX-1 function. However, a clear mechanism for how perforation of the phagosome by ESX-1/PDIM facilitates pathogenesis is lacking. The current study by Hinman et al. centers around a detailed comparison of the transcriptional responses of macrophages to infection with wild-type, ESX-1 mutant, and PDIM mutant Mtb. Through transcriptional profiling they identify a set of genes that have increased expression levels in macrophages infected with the mutant strains. These genes are part of a module of genes that are expressed late after TLR2 stimulation. The authors propose that this late gene expression results from TLR2 signaling from the phagosomal/endosomal compartment, and that this endosomal/phagosomal signaling is enhanced by endosomal acidification. The authors propose that these genes have increased expression levels in infections with mutants lacking ESX because these mutants exist in a compartment with lower phagosomal pH, and suggest that blunting the expression of host response genes is a pathogenic strategy. The transcriptional profiling work is sound, the defined clusters TLR signaling are clear, and the manuscript itself is well written. However, it is unclear the observations reported in the study have any real significance for pathogenesis or the regulation of TLR2 signaling.

1. The idea that TLR2 might signal from the endosome/phagosome has been previously proposed. However, rigorous proof is lacking. This study does not clearly add weight to the idea that TLR2 signals from the endosome/phagosome as Dynasore data alone is not sufficient. Furthermore, even if TLR2 does signal from the endosome, this study does not demonstrate that compartment specific signaling of TLR2 has important functional consequences.

2. The findings do not advance our understanding of ESX-1 function or Mtb pathogenesis. Although some of the genes identified as being suppressed by ESX-1 have links to pathogenesis, it is not clear that the differences in expression observed in this study have any functional relevance for infection.

3. The authors claim that the perforation of the membrane results in the blunting of phagosomal TLR2 signaling, however all of the data is correlative. PDIM and ESX-1 mutants likely participate int he same pathway of membrane damage and do not represent independent confirmation. Further, ESX-1 mutants are already known to fail to arrest phagosome maturation and reside within an acidified compartment. The authors do not clearly establish that it is the physical perforation itself that results in the altered pH, as they propose. Furthermore, most attenuated mutants would be assumed to traffic to the lysosome, independent of membrane perforation. Thus one might predict that any attenuated mutant – or even heat killed bacteria – would result in enhanced expression of the same group of genes identified as highly expressed in this study.

*Reviewer #3:*

In this article, Hinman and al. describe a novel *Mycobacterium tuberculosis* (Mtb) evasion mechanism whereby the bacterium's virulence factors ESX-1 and PDIM undermine the host endosome-specific TLR2 activation through endosome membrane damage and limitation of phagosome acidification. These findings are of high importance as they may inform host-directed therapeutic approach aimed at enhancing TLR2 activation, thus more effective inflammatory response against Mtb. In general the manuscript is well written, the literature cited is very relevant, the methodology used was appropriate. The results are well presented and discussed; figures and tables are clear.

1. The authors are requested to justify for the choice of the C57Bl6 mouse background (prone to mount proinflammatory response) for these experiments. would they observe the same two-component TLR2-dependent response in Mtb-challenged BalbC mouse background (less prone to mount proinflammatory response)?

2. The authors used different MOI in different experiments (figures 2 and 3) they stated in the methodology section (page 11) that the MOI used for each time point was selected to maximize signal while minimizing infection-associated cell death. It is also known that different MOI may result in different host trancriptome change. How did the authors correct for that?

---

## [Author Response]

[Editors’ note: the authors resubmitted a revised version of the paper for consideration. What follows is the authors’ response to the first round of review.]

Reviewer #1:The article entitled "Mycobacterium tuberculosis canonical virulence factors interfere with compartment-specific TLR-2 signaling" by Hinman et al. provides insight into the TRL2 mediated two-component immune response acting through two clusters of genes at an early and late stage of infection in vitro. The authors focus on the late stage TLR2 signaling cascade, which they show is dependent on vesicle acidification. In general, the experiments were rigorously performed and well-presented.

We appreciate the comment that our experiments were in general rigorously performed and well-presented.

A major concern is the variation in MOI that was used throughout the manuscript. The authors mention that "the MOI used for each time point was selected to maximize signal while minimizing infection-associated cell death", but how do the authors correct for different MOIs across experiments? For example, in Figure-1, MOI 2 of infection was used for RNAseq whereas MOI 10 was used for qPCR analysis. The authors should mention the reason behind switching from a lower dose infection (MOI 2) to a higher dose infection (MOI 10). It would be interesting to see an MOI dependent expression pattern of Saa3, Lcn2, and Marco at 16h post-infection in figure 1. As another example, TNF expression was studied at different time points with MOI 10 in figure 2C which suggested that at 8hr post-infection, the expression of TNF reaches its peak. But then the authors reduced the MOI to 5 and time point to 6hr post-infection for the TNF expression analysis in Figure 3D.

We appreciate the reviewer’s point that experimental demonstration of the early and late responses across MOIs would offer better context for interpreting our results. As suggested by the reviewer, we have included profiling of late component genes over time at MOIs of 2:1 (Figure 2A) and 10:1 (Figure 2B) and early component genes over time at MOIs of 2:1 (Figure 3C) and 10:1 (Figure 3D). We have included the closer time course comparison of the early component at MOIs of 5:1 and 10:1 present in the initial manuscript (Figure 3—figure supplement 1A). As described in the text (lines 141-143 and 179-186), for all strains the magnitude of gene expression is enhanced at higher MOI, but the relationships between responses to wild-type and PDIM- or ESX-1-mutant Mtb remain the same.

For Figure 4, the authors could show a time-dependent pH measurement for each Mtb strain (eg. 12h, 24h, 48h) to more precisely coordinate the timing of acidification with the second wave of TLR signaling.

We appreciate this suggestion for characterizing the kinetics of phagosome acidification in our model, and have added data for all strains 6 hours, 12 hours, and 24 hours post-infection (Figure 6A). This data demonstrates that differential phagosome acidification occurs by the time we begin to see differential gene expression (12 hours).

It would be nice if the authors provide a representative 5 pixel expanded microscopy image as they mentioned in Material and Methods in the supplementary data for the phagosomal pH analysis of each Mtb strain.

We have added representative microscopy images (Figure 6—figure supplement 1A) and images demonstrating the CellProfiler identification of GFP-labeled bacteria with 5 pixel expansion to detect pHrodo (Figure 6—figure supplement 1B).

Reviewer #2:The ESX-1 type VII secretion system is critical for virulence of M. tuberculosis and has been proposed to function in part by perforation of phagosomal membrane. In a previous study, the corresponding author found that the complex lipid PDIM is linked to ESX-1 function. However, a clear mechanism for how perforation of the phagosome by ESX-1/PDIM facilitates pathogenesis is lacking. The current study by Hinman et al. centers around a detailed comparison of the transcriptional responses of macrophages to infection with wild-type, ESX-1 mutant, and PDIM mutant Mtb. Through transcriptional profiling they identify a set of genes that have increased expression levels in macrophages infected with the mutant strains. These genes are part of a module of genes that are expressed late after TLR2 stimulation. The authors propose that this late gene expression results from TLR2 signaling from the phagosomal/endosomal compartment, and that this endosomal/phagosomal signaling is enhanced by endosomal acidification. The authors propose that these genes have increased expression levels in infections with mutants lacking ESX because these mutants exist in a compartment with lower phagosomal pH, and suggest that blunting the expression of host response genes is a pathogenic strategy. The transcriptional profiling work is sound, the defined clusters TLR signaling are clear, and the manuscript itself is well written. However, it is unclear the observations reported in the study have any real significance for pathogenesis or the regulation of TLR2 signaling.

We thank the reviewer for the comments that “the transcriptional profiling work is sound, the defined clusters TLR signaling are clear, and the manuscript itself is well written”.

1. The idea that TLR2 might signal from the endosome/phagosome has been previously proposed. However, rigorous proof is lacking. This study does not clearly add weight to the idea that TLR2 signals from the endosome/phagosome as Dynasore data alone is not sufficient.

We agree with the reviewer’s comment that additional data demonstrating the relevance of uptake would more compellingly support the conclusion that endosomal uptake is required for the late component of the TLR2 response. In new experiments, we additionally tested the impact of the actin polymerization inhibitor cytochalasin D, which has been used extensively to distinguish surface- from endosome-initiated innate immune signaling. This data is included as new Figure 5E. While a genetic approach could offer an additional window into the question, this approach is predicated upon identifying an adapter uniquely required for the response of interest. We tested all known adapters, but did not identify an adapter required for the late response to TLR2. This data testing known adapters is included as new Figure 4—figure supplement 1C-D. We have softened the language to indicate that the data is suggestive of/consistent with (rather than iron-clad proof of) compartment-specific signaling, and have changed the title of the manuscript.

Furthermore, even if TLR2 does signal from the endosome, this study does not demonstrate that compartment specific signaling of TLR2 has important functional consequences.

We agree that understanding whether there is relevance for pathogenesis is important for the significance of our findings. To address this question, we studied the relevance of the TLR2/PDIM and TLR/ESX-1 interaction for two key outcomes of infection in macrophages- macrophage cell death and intracellular bacterial growth. We find that the relative attenuation of PDIM and ESX-1 mutants for both phenotypes is partially reversed in Tlr2-/- BMDM. This data is shown in new figures 7A-C, and is discussed in the new Results section “PDIM and ESX-1 modulate TLR2-dependent infection outcomes in macrophages”, lines 335-367. We additionally studied the relevance of the TLR2/PDIM interaction for bacterial growth and histopathology in mice and found that the relative attenuation of the PDIM mutant is partially reversed in Tlr2-/- mice. This data is shown in new figures 7D and Figure 7—figure supplement 1, and discussed in the new Results section “PDIM and ESX-1 modulate TLR2-dependent infection outcomes in mice”, lines 369-401.

2. The findings do not advance our understanding of ESX-1 function or Mtb pathogenesis. Although some of the genes identified as being suppressed by ESX-1 have links to pathogenesis, it is not clear that the differences in expression observed in this study have any functional relevance for infection.

As noted above, we agree that understanding the relevance of our findings in the context of infection is important, and we have added experiments in macrophages and mice to test this relevance.

3. The authors claim that the perforation of the membrane results in the blunting of phagosomal TLR2 signaling, however all of the data is correlative. PDIM and ESX-1 mutants likely participate int he same pathway of membrane damage and do not represent independent confirmation. Further, ESX-1 mutants are already known to fail to arrest phagosome maturation and reside within an acidified compartment. The authors do not clearly establish that it is the physical perforation itself that results in the altered pH, as they propose.

To address this point with an orthogonal experimental approach, we attempted to use Llome, an agent that induces sterile damage. Unfortunately, we found that all concentrations that induce membrane damage kill BMDM by 8 hours post-infection, limiting our ability to profile a late transcriptional response. We have thus softened the language of our claims to indicate that our results would be consistent with Mtb-mediated membrane damage

Furthermore, most attenuated mutants would be assumed to traffic to the lysosome, independent of membrane perforation. Thus, one might predict that any attenuated mutant – or even heat killed bacteria – would result in enhanced expression of the same group of genes identified as highly expressed in this study.

To address this point, we obtained a published, well-characterized Mtb mutant highly attenuated for growth in macrophages. We confirmed the attenuation phenotype in BMDM (new Figure 2—figure supplement 1B) and tested whether BMDM infection with this mutant resulted in an enhanced late TLR2 response. We found that it did not; this data is included as new Figure 2—figure supplement 1C.

Reviewer #3:In this article, Hinman and al. describe a novel Mycobacterium tuberculosis (Mtb) evasion mechanism whereby the bacterium’s virulence factors ESX-1 and PDIM undermine the host endosome-specific TLR2 activation through endosome membrane damage and limitation of phagosome acidification. These findings are of high importance as they may inform host-directed therapeutic approach aimed at enhancing TLR2 activation, thus more effective inflammatory response against Mtb. In general the manuscript is well written, the literature cited is very relevant, the methodology used was appropriate. The results are well presented and discussed; figures and tables are clear.

We appreciate the positive comments of the reviewer.

1. The authors are requested to justify for the choice of the C57Bl6 mouse background (prone to mount proinflammatory response) for these experiments. would they observe the same two-component TLR2-dependent response in Mtb-challenged BalbC mouse background (less prone to mount proinflammatory response)?

We appreciate this point. C57BL/6J mice are a well-established model for Mtb infection, and many knockout mice lacking individual innate immune signaling components are available in the C57BL/6J background, facilitating the types of comparative studies we performed with *Tlr4*-/- and *Tlr2*-/- BMDM in this manuscript. These were the factors that drove our initial choice of C5BL/6J BMDM as a model. To experimentally address the question of whether BMDM from mice with a less inflammatory baseline state would respond similarly, we tested BALB/c BMDM for induction of the late TLR2-dependent response, and found that it was similarly induced and enhanced in response to PDIM and ESX-1 mutant-Mtb. This data is included as new Figure 2—figure supplement 1A.

2. The authors used different MOI in different experiments (figures 2 and 3) they stated in the methodology section (page 11) that the MOI used for each time point was selected to maximize signal while minimizing infection-associated cell death. It is also known that different MOI may result in different host trancriptome change. How did the authors correct for that?

We appreciate this comment. As noted in the response to Reviewer 1, we have included profiling of late component genes over time at MOIs of 2:1 (Figure 2A) and 10:1 (Figure 2B) and early component genes over time at MOIs of 2:1 (Figure 3C) and 10:1 (Figure 3D). We have additionally included a closer time course comparison of the early component at MOIs of 5:1 and 10:1 (Figure 3—figure supplement 1A). As described in the text (lines 141-143 and 179-186), for all strains the magnitude of gene expression is enhanced at higher MOI, but the relationships between responses to wild-type and PDIM- or ESX-1-mutant Mtb remain the same.